# A Shape Approximation for Medical Imaging Data

**DOI:** 10.3390/s20205879

**Published:** 2020-10-17

**Authors:** Shih-Feng Huang, Yung-Hsuan Wen, Chi-Hsiang Chu, Chien-Chin Hsu

**Affiliations:** 1Department of Applied Mathematics, National University of Kaohsiung, Kaohsiung 811, Taiwan; 2Institute of Statistics, National University of Kaohsiung, Kaohsiung 811, Taiwan; m1074405@mail.nuk.edu.tw; 3Department of Statistics, National Cheng Kung University, Tainan 701, Taiwan; 10902042@gs.ncku.edu.tw; 4Department of Nuclear Medicine, Kaohsiung Chang Gung Memorial Hospital, Chang Gung University College of Medicine, Kaohsiung 833, Taiwan; cchsu@cgmh.org.tw

**Keywords:** imaging data, Parkinson’s disease, PSO algorithm, shape equation

## Abstract

This study proposes a shape approximation approach to portray the regions of interest (ROI) from medical imaging data. An effective algorithm to achieve an optimal approximation is proposed based on the framework of Particle Swarm Optimization. The convergence of the proposed algorithm is derived under mild assumptions on the selected family of shape equations. The issue of detecting Parkinson’s disease (PD) based on the Tc-99m TRODAT-1 brain SPECT/CT images of 634 subjects, with 305 female and an average age of 68.3 years old from Kaohsiung Chang Gung Memorial Hospital, Taiwan, is employed to demonstrate the proposed procedure by fitting optimal ellipse and cashew-shaped equations in the 2D and 3D spaces, respectively. According to the visual interpretation of 3 experienced board-certified nuclear medicine physicians, 256 subjects are determined to be abnormal, 77 subjects are potentially abnormal, 174 are normal, and 127 are nearly normal. The coefficients of the ellipse and cashew-shaped equations, together with some well-known features of PD existing in the literature, are employed to learn PD classifiers under various machine learning approaches. A repeated hold-out with 100 rounds of 5-fold cross-validation and stratified sampling scheme is adopted to investigate the classification performances of different machine learning methods and different sets of features. The empirical results reveal that our method obtains 0.88 ± 0.04 classification accuracy, 0.87 ± 0.06 sensitivity, and 0.88 ± 0.08 specificity for test data when including the coefficients of the ellipse and cashew-shaped equations. Our findings indicate that more constructive and useful features can be extracted from proper mathematical representations of the 2D and 3D shapes for a specific ROI in medical imaging data, which shows their potential for improving the accuracy of automated PD identification.

## 1. Introduction

In the medical field, computed tomography (CT), magnetic resonance imaging (MRI), positron emission tomography (PET), and single-photon emission computed tomography (SPECT) have been used for preventive medicine, screening for disease, and confirming a diagnosis. These techniques are capable of providing the three-dimensional (3D) structure of a region of interest (ROI) by a series of two-dimensional (2D) images sequentially ordered in time.

In clinical practice, doctors use their domain knowledge to select a subset of the 2D images from the series. The selected images are superimposed into a single 2D image, which is called a 2D-combined image hereafter, for disease diagnosis. For example, Prashanth et al. [1], Taylor et al. [2], Oliveira et al. [3], Nicastro et al. [4], and Iwabuchi et al. [5] constructed classifiers for identifying Parkinson’s disease (PD) by extracting features from 2D-combined SPECT images. However, this approach may lose some useful information or features for classification since a 2D-combined image is a projection of the original 3D structure on a specific 2D space. Recently, some methodologies have been proposed to draw information from medical 3D images. For instance, Ziegler et al. [6] used multispectral structural magnetic resonance imaging (MRI) tools to measure substantia nigra volume loss before basal forebrain degeneration in early PD. Palumbo et al. [7] proposed a volumetric 3D region of interest (ROI) of putamen and caudate nucleus by BasGan software. Prashanth et al. [8] proposed to draw ‘indirect 3D features’ by building a regression surface of an ROI based on the 2D-combined image. Hsu et al. [9] found that the classification performance for identifying PD is improved by calculating the volumes of an ROI in 3D space. Cheng et al. [10] adopted the 3D features such as volume, surface area, and diameter to describe the geometric characteristics of the volumes of interest to assist the diagnosis of idiopathic PD. Xu et al. [11] assessed the longitudinal volume change of different hippocampal subfields in PD patients with and without cognitive decline using 3D T1-weighted MRI.

Most of the aforementioned studies of drawing 3D features focused on computing the volumes of the ROI and applying the quantities derived from the volumes of the ROI to PD identification [6,7,9,10,11]. Although the geometric features such as volume and the volume-related quantities are valuable proxies representing the 3D shapes of the ROI, they still cannot fully capture the desired shapes since different shapes of the ROI could have the same volumes or volume-related quantities. This fact could restrict the classification performances of these volume-based methods. In addition, the 3D shapes of the ROI did affect nuclear medicine physicians’ judgment in practice. In visual interpretation, the normal striatal shape looks cashew-shaped in 3D space. The common pattern of abnormalities in PD is usually a decrease uptake in the putamen (comma-shaped) initially and progressing anteriorly to caudate nucleus over time. The striatal shape becomes oval-shaped gradually with disease progression. Consequently, if there is a way to capture the 3D shapes of the ROI for both normal and abnormal subjects more specifically and directly than just computing the volumes of the ROI, we might have the chance to obtain more useful shape features for classification. To fill this gap and unlike the indirect approach of [8] to obtain ‘3D features’, we propose to directly characterize the 3D shape of an ROI by a suitable mathematical representation. The coefficients of the fitted equations are then adopted to learn classifiers to reflect the influence of the 3D shapes of the ROI. To the best of our knowledge, this is the first work using a family of mathematical equations to portray the 3D shapes of the ROI for normal and abnormal subjects simultaneously.

Throughout this paper, we adopt the issue of identifying PD to illustrate our approach. Parkinsonian syndromes (PS) are a group of movement disorders characterized by tremor, bradykinesia, and rigidity. PD is the most common neurodegenerative movement disorder and accounts for 75% of all PS. PD with early death of dopaminergic neurons in the substantia nigra pars compacta and dopamine deficiency within the basal ganglia leads to classical parkinsonian motor symptoms [12,13]. It is important to differentiate between PS caused by nigrostriatal dopaminergic degeneration and nondegenerative causes of parkinsonism such as essential tremor, vascular parkinsonism, psychogenic parkinsonism, or drug-induced parkinsonism. Dopamine transporter (DAT) imaging can detect presynaptic dopamine neuronal dysfunction and is useful in the differentiation between conditions with and without presynaptic dopaminergic deficit. Currently, SPECT radioligands for DAT, such as I-123 FP-CIT and Tc-99m TRODAT-1, have been available for daily clinical practice [14,15]. In practice, the healthy cases and abnormal cases have different appearances of uptakes of the ROI in 2D-combined and 3D imaging data. For example, a healthy subject has comma-shaped uptakes in his/her 2D-combined SPECT image while a subject with PD has dotted-shaped uptakes. In addition, a healthy subject has cashew-shaped uptakes around his/her striatum in 3D space while a subject with PD has sphere-shaped uptakes [8]. As a result, we propose to depict the appearances of uptakes of the ROI by proper mathematical formulas for the corresponding 2D and 3D shapes, which create a possibility to identify the differences between healthy and abnormal subjects by the associated coefficients contained in the mathematical representations. Specifically, for the issue of identifying PD, we fit ellipses for the uptakes in the 2D-combined image and cashew-shaped 3D equation for the uptakes in the 3D space for each subject under the framework of minimizing an associated loss function. Nevertheless, when conducting these two shape fittings, we have to face multidimensional optimization problems, which usually have no closed-form solution and one can only rely on numerical approximations [16].

In this study, we adopt the Particle Swarm Optimization (PSO) algorithm proposed by [17] to deal with the considered shape fittings. The PSO algorithm is inspired by the behavior of social organisms in groups, such as bird and fish schooling or ant colonies. It has been applied to numerous areas and has been shown to be capable of solving multidimensional optimization problems effectively (see [18,19,20,21] and the references therein). Another challenge encountered in the shape fitting process is to compute the Euclidean distance (also called L2 distance) between an observation and a given shape equation in $R^{2}$ or $R^{3}$ space. Morera et al. [22] proposed to calculate the L2 distance from a point to a quadratic surface by a partial differential equation (PDE) procedure. However, their PDE procedure is difficult to be extended to higher-order surfaces like the cashew-shaped equations considered in this study. To overcome this challenge, we propose a “flashlight-searching” (FS) algorithm to compute the distance for higher-order cases. The FS algorithm is shown to be capable of producing accurate results for the quadratic examples in [22]. In addition, under some mild assumptions on the selected family of shape equations, the convergence of the proposed FS algorithm is derived, which reveals the effectiveness and easy-for-implementation of the PSO-FS algorithm. Once conducting the proposed shape fitting procedure, the coefficients of the optimal 2D ellipse and cashew-shaped 3D equation are employed to learn classifiers for PD.

Between January 2017 and May 2018, 634 patients who underwent Tc-99m TRODAT-1 brain SPECT/CT in the Department of Nuclear Medicine, Kaohsiung Chang Gung Memorial Hospital, Taiwan, were enrolled in this retrospective study. The Chang Gung Medical Foundation Institutional Review Board approved this retrospective study and waived the requirement for written informed consent. The TRODAT SPECT images were interpreted by three experienced-board-certified nuclear medicine physicians, and were assigned into normal or abnormal accordingly. Six popular classification methods such as logistic regression (LR), linear discriminant analysis (LDA), naive Bayes (NB), random forests (RF), support vector machine (SVM), and extreme gradient boost (XGB) are adopted to learn classifiers for PD using the coefficients contained in the 2D and 3D shape-fitting equations. The 6 classifiers are usually adopted in many comparison studies for medical data [3,9,23,24,25]. The numerical results reveal that these coefficients did improve the classification accuracies for most of the considered classifiers. Beyond the 6 machine learning classifiers considered in this study, deep learning (DL) is one of the hottest technology applied to classification problem in many fields. Recently, Dolz et al. [26] proposed a 3D and fully convolutional neural network method for subcortical brain structure segmentation in MRI, and yielded satisfactory segmentations. The objective of this study is to show that suitable shape equations of the ROI are useful for classification. Intuitively, the shape equations of the ROI will be more precise if better segmentations are used. Therefore, it will be interesting to construct shape equations based on Dolz et al. [26]’s segmentations. Nevertheless, it is beyond the scope of this work, and we leave it to our future study.

The remainder of this paper is organized as follows. Section 2 describes the proposed criteria for determining suitable shape equations to approximate the regions where we are interested in. Section 3 introduces an easy-to-implement algorithm for obtaining the optimal solutions of the considered criteria. The convergence of the proposed algorithm is also derived. The numerical results of applying the proposed approach to PD imaging data are presented in Section 4. Discussions are presented in Section 5. Technical calculations and proofs are given in the Appendix A.

## 2. Optimization Criteria

In this section, we introduce the proposed criteria for shape fitting in 2D and 3D spaces. Shape analysis and surface fitting are useful and promising methods for extracting features, which can be used to develop diagnostic models [8,27,28,29]. To construct a classifier for PD identification, Prashanth et al. [8] noticed that the shapes of uptakes in the 2D-combined images are quite different between normal and PD subjects. They employed a pair of ellipses to describe the ROI for a 2D-combined PD image, and many features are then calculated from the fitted ellipse equations. For example, the lengths of major and minor axes, the eccentricity, orientation, roundness, area, etc. of an ellipse can be computed directly from the mathematical representation of the fitted ellipses. Moreover, they proposed to use the following polynomial equation of order 3 to extract 3D features from the 2D-combined images:(1)$$f\left(x,y\right)=p_{00}+p_{10}x+p_{01}y+p_{20}x^{2}+p_{11}xy+p_{02}y^{2}+p_{30}x^{3}+p_{21}x^{2}y+p_{12}xy^{2}+p_{03}y^{3},$$
where f(x,y) is the grayscale with respect to the point (x,y) on the 2D plane and the coefficients $p_{ij},i,j\in \left\{0,1,2,3\right\}$ are estimated by minimizing the L2 loss. Figure 1 presents the fitting results of normal and PD subjects by using Prashanth et al. [8]’s approach with (Equation 1). Note that for demonstration, we present the color version of the SPECT images throughout this study. Prashanth et al. [8] used several characteristics calculated from the fitted ellipses and the coefficients in (Equation 1) as features to learn an SVM classifier. Their numerical results reveal that the classification accuracy for PD can be effectively improved. One interesting point should be noticed that Prashanth et al. [8] did not construct a model for the real shapes of uptakes in the 3D space directly, but established a regression model for the grayscale surface by using the 2D-combined image of each subject solely. This motivates us to investigate that whether we could obtain more useful features extracted from modeling the real shapes of uptakes in the 3D space according to the series of 2D scanned images of each subject. The details are illustrated in the following.

Let $p_{i}$, i=1,2,…,n denote the boundary points of the ROI. In Figure 2, the boundary points with grayscale value greater than 0.6 of two images are marked in black, where the left one is an image of a normal subject and the right one is an image of a subject with PD. In addition, let Ψ(·;θ)=0 denote an equation which is suitable to well depict the shape of the ROI, where θ are unknown parameters. For example, Figure 2 shows that ellipse equations are suitable to depict the shapes of the ROI shown in a 2D-combined image and Figure 3 reveals that cashew-shaped equations are suitable to depict the structures of the ROI for a normal case or a PD case in 3D space. Intuitively, normal subjects should have similar 2D and 3D shapes while the subjects with PD should have different shapes from normal subjects. Potentially, the coefficient θ and the properties of the equation Ψ(·;θ)=0 could provide useful information for separating normal and abnormal cases. Nevertheless, the location of the center and the rotation angles of the surface of the uptakes for each subject (even for normal subjects) would have slight differences, which increase the complexity of fitting a suitable surface equation for the boundary points in practical implementation. In this study, we adopt the image data of PD to demonstrate the performances of the proposed method. As shown in Figure 2 and Figure 3, the ellipse equations and cashew-shaped equations are selected to depict the ROI in 2D and 3D spaces, respectively. The details of the two families of equations for including the considerations mentioned above and the corresponding optimization criteria for 2D and 3D cases are introduced in the following.

For 2D surface fitting, an ellipse equation $\Psi_{2D}\left(x;\theta_{2D}\right)=0$ is employed, where
(2)$$\begin{array}{cc} \Psi_{2D}\left(x;\theta_{2D}\right)= & Ax^{2}+2Bxy+Cy^{2}+Dx+Ey+F, \end{array}$$
in which x=(x,y), $B^{2}<AC$ and $\theta_{2D}=\left(A,B,C,D,E,F\right)\in R^{6}$. Let $H\left(\theta_{2D}\right)=\left\{x:\Psi_{2D}\left(x;\theta_{2D}\right)=0\right\}$ denote the set of points on the ellipse. An optimization criterion for finding a suitable $H(\theta_{2D})$, abbreviated by $H_{2D}$, for the observed boundary points $p_{i}$, i=1,…,n, in a 2D-combined image:(3)$$\begin{array}{c} \theta_{2D}^{*}=\arg\min_{\theta_{2D}}\sum_{i=1}^{n}d\left(p_{i},H_{2D}\right), \end{array}$$
where $d\left(p_{i},H_{2D}\right)=\min_{x\in H_{2D}}d\left(p_{i},x\right),$ and $d\left(p_{i},x\right)=∥p_{i}-x∥$ denote the Euclidean distance between $p_{i}$ and x.

For 3D surface fitting, a cashew-shaped equation is defined as $\Psi_{3D}^{0}\left(x;\theta_{3D}\right)=0$, where
(4)$$\Psi_{3D}^{0}\left(x;\theta_{3D}^{0}\right)={\left(1.5x^{2}+y^{2}+z^{2}+a^{2}-b^{2}\right)}^{2}-4a^{2}\left({\left(x-c\right)}^{2}+{\left(y-d\right)}^{2}\right),$$
in which x=(x,y,z) and $\theta_{3D}^{0}=\left(a,b,c,d\right)\in {\left[0,1\right]}^{4}$. In order to consider the translation, rotation, and scaling of the equation, we do the following:Rotate x by the following rotation matrix *M* with three radians $\theta_{x},\theta_{y},\theta_{z}\in {\left[0,2\pi \right]}^{3}$:
$$M=\left(\begin{array}{ccc} 1 & 0 & 0\\ 0 & \cos(\theta_{x}) & \sin(\theta_{x})\\ 0 & -\sin(\theta_{x}) & \cos(\theta_{x}) \end{array}\right)\times \left(\begin{array}{ccc} \cos(\theta_{y}) & 0 & -\sin(\theta_{y})\\ 0 & 1 & 0\\ \sin(\theta_{y}) & 0 & \cos(\theta_{y}) \end{array}\right)\times \left(\begin{array}{ccc} \cos(\theta_{z}) & \sin(\theta_{z}) & 0\\ -\sin(\theta_{z}) & \cos(\theta_{z}) & 0\\ 0 & 0 & 1 \end{array}\right)$$Furthermore, by translating and scaling x by $t=\left(t_{x},t_{y},t_{z}\right)\in R^{3}$ and s>0, respectively, the general form of a cashew-shaped equation is represented by
(5)$$\Psi_{3D}\left(x;\theta_{3D}\right)=\Psi_{3D}^{0}\left(sM\left(x-t\right);\theta_{3D}^{0}\right),$$
where $\theta_{3D}=\left(a,b,c,d,s,t_{x},t_{y},t_{z},\theta_{x},\theta_{y},\theta_{z}\right)$ and $\Psi_{3D}^{0}\left(\cdot \right)$ is defined in (Equation 4).

Based on the representation shown in (Equation 5), let $H\left(\theta_{3D}\right)=\left\{x:\Psi_{3D}\left(x;\theta_{3D}\right)=0\right\}$ denote the set of points on the cashew-shaped. Similar to (Equation 3), an optimization criterion for finding a suitable $H(\theta_{3D})$, abbreviated by $H_{3D}$, for the observed boundary points $p_{i}$, i=1,…,n, in 3D space:(6)$$\theta_{3D}^{*}=\arg\min_{\theta_{3D}}\sum_{i=1}^{n}d\left(p_{i},H_{3D}\right),$$
where $d\left(p_{i},H_{3D}\right)=\min_{x\in H_{3D}}d(p_{i},x)$.

The idea for determining optimal shape equations through the criteria shown in Equations (3) and (6) for 2D and 3D cases, respectively, is based on minimizing the L2 distance of the observed points to an ellipse or a cashew-shaped equation, which is similar to the classical least-squares (LS) regression. However, the classical LS cannot be applied directly for the equations $\Psi_{2D}=0$ and $\Psi_{3D}=0$. For example, the variable *y* in the equation $\Psi_{2D}=0$ is not a function of *x*, which makes the shape approximation problem different from classical regression. Therefore, we propose an algorithm to find the optimal solutions of (Equation 3) and (Equation 6), and the details are introduced in the next section.

## 3. The Proposed Algorithm

For the boundary points $p_{i}$, i=1,…,n, and a determined family of surface equation Ψ(·;θ)=0, we propose to use an optimal surface satisfying $\Psi (\cdot ;\theta^{*})=0$ which has the minimal L2 distance to the boundary points $p_{i},i=1,\ldots ,n$, to approximate the shapes of the uptakes of the ROI as shown in (Equation 3) and (Equation 6). Since the optimal solution $\theta^{*}$ is a high-dimensional vector and there is usually no closed-form representation for $\theta^{*}$, a PSO framework is employed to obtain $\theta^{*}$ and an effective scheme, called “flashlight-searching” (FS), is proposed to calculate the distance from a point to a surface in each PSO iteration. Herein, the proposed algorithm is abbreviated by “PSO-FS”. In the following, we briefly introduce the PSO procedure.

Kennedy et al. [17] proposed an optimization algorithm, called PSO, which searches the optimal solution through particles and emulates the interaction of information sharing among particles iteratively. In each iteration, the updated search position of each particle is determined by the best position in its historical trajectory and the best position found in the trajectories of all particles. Specifically, we denote the position of the *i*-th particle in $R^{d}$ by a *d*-dimensional vector $x_{i}$ and denote its velocity vector by $v_{i}$, i=1,…,n. In the (t+1)-th iteration, each particle changes its position according to the new velocity. That is, the position $x_{i}^{t+1}$ is defined as:(7)$$x_{i}^{t+1}=x_{i}^{t}+v_{i}^{t+1}$$
with the new velocity $v_{i}^{t+1}$ defined by
(8)$$v_{i}^{t+1}=\omega v_{i}^{t}+c_{1}r_{1}\left(P_{i}^{t}-x_{i}^{t}\right)+c_{2}r_{2}\left(G^{t}-x_{i}^{t}\right),$$
where $P_{i}^{t}$ and $G^{t}$ denote the best position of the *i*-th particle and the best position of all particles up to the *t*-th iteration, respectively. The parameters ω, $\left(c_{1},c_{2}\right)$ and $\left(r_{1},r_{2}\right)$ are inertia weight, two positive constants and two random parameters within [0,1], respectively. The PSO algorithm is repeated until a predetermined stopping criterion is reached or the change rate of the particles approaches zero.

For example, suppose that the objective is to find the minimum of a surface $f:R^{2}\to R$, where the contour plots of *f* with the paths of 5 particles for the first 4 iterations of PSO are shown in Figure 4. In the four panels of Figure 4, the solid points denote the locations of the 5 particles in each iteration and the gray points combined with dashed lines denote the corresponding path of each particle. For each particle, the red arrow denotes the velocity of the *i*-th particle $v_{i}^{t+1}$ in the *t*-th iteration, and the black, green, and yellow dotted-arrows denote the 3 components, $v_{i}^{t}$, $\left(P_{i}^{t}-x_{i}^{t}\right)$, and $\left(G^{t}-x_{i}^{t}\right)$, respectively, on the right-hand side of (Equation 8) to obtain $v_{i}^{t+1}$. In addition, the green circle represents $P_{i}^{t}$ and the particle with red cross represents $G^{t}$. From Figure 4, one can find that the particles did share their information with others to determine their searching directions in each iteration and all of them tend to approach the point with minimal *f* gradually.

In this study, we employ the PSO procedure to obtain the optimal solutions of the criteria (Equation 3) and (Equation 6) in 2D and 3D spaces, respectively. For example, to proceed with the PSO procedure for searching $\theta_{2D}^{*}$ defined in (Equation 3), the vector $x_{i}^{t}$ defined in (Equation 7) is set to be $x_{i}^{t}=\theta_{2D,i}^{t}$, where $\theta_{2D,i}^{t}$ denotes the position of the *i*-th particle at the *t*-th PSO iteration. In addition, the $P_{i}^{t}$ and $G^{t}$ defined in (Equation 8) are computed from $\Psi_{2D}\left(\cdot \right)$ defined in (Equation 2).

Nevertheless, to calculate (Equation 3) and (Equation 6) in each PSO iteration, we have to calculate $d(p_{i},H)$, which usually does not have a closed-form representation and a numerical method can help to compute the distance. For example, Morera et al. [22] proposed a numerical method under the framework of PDE to compute the L2 distance from a point to a quadric surface. However, their approach heavily relies on the properties of quadric equations when handling the corresponding PDEs and is not trivial to be extended to the cases of higher order surfaces, like the cashew-shaped equations considered in this study. Therefore, we propose an easy-to-implement FS algorithm instead to overcome the difficulty. The details of the FS algorithm are illustrated in the following.

For any given $p\in R^{d}$, d≥2, let $D_{p}\subseteq H$ denote a set satisfying
(9)$$D_{p}=\left\{x^{*}\in H:d\left(p,x^{*}\right)=d\left(p,H\right)\right\},$$
which allows the case of more than one $x^{*}\in H$ such that $d\left(p,x^{*}\right)=d\left(p,H\right)$. Further, let ${\bar{B}}_{\epsilon }\left(x^{*}\right)=\{x:||x-x^{*}\left|\right|\leq \epsilon ,x^{*}\in D_{p}\}$ denote a closed ball of radius ϵ>0 centered at an $x^{*}\in D_{p}$, and $J=\{{⋃}_{x^{*}\in D_{p}}{\bar{B}}_{\epsilon }\left(x^{*}\right)\}\cap H$ denote the arc of *H* associated with ${⋃}_{x^{*}\in D_{p}}{\bar{B}}_{\epsilon }\left(x^{*}\right)$. Since *J* is allowed to be a disconnected set, let $\left\{J_{i},i=1,2,\ldots \right\}$ denote a partition of *J*, where ${⋃}_{i=1,2,\ldots }J_{i}=J$, $J_{i}\cap J_{k}=$ for i≠k, and each $J_{i}$ is a connected set. A schematic diagram of these notations is given in Figure 5 for the special case of $D_{p}$ containing only one $x^{*}$. The FS algorithm for computing $d(p_{i},H)$ proceeds as follows.

Set an r>0 such that for all $y\in {\bar{B}}_{r}\left(p\right)$, we have Ψ(y)>0orΨ(y)<0, where ${\bar{B}}_{r}\left(p\right)$ is defined similarly to ${\bar{B}}_{\epsilon }\left(x^{*}\right)$. Let $x_{0}\in H$ denote the initial point for estimating d(p,H) and $x_{0}$ can be obtained by letting
(10)$$x_{0}=c\left(y_{r}-p\right)+p,\;c\in R,$$
where $y_{r}=\arg\min_{y}\left\{\Psi \left(y\right):y\in {\bar{B}}_{r}^{b}\left(p\right)\right\}$ and ${\bar{B}}_{r}^{b}\left(p\right)$ denotes the set of boundary points of ${\bar{B}}_{r}\left(p\right)$.Let τ be a predetermined tolerance, and set m=0.(i)Compute the tangent plane to Ψ at $x_{m}$ and denote it by $\Gamma_{m}$. Let $\left\{x_{m,j},j=1,\ldots ,k\right\}$ be a set of points on $\Gamma_{m}$ and $\left|\right|x_{m,j}-x_{m}\left|\right|=\delta$, where δ is a predetermined positive constant.(ii)Let ${\tilde{x}}_{m,j}\in J$ and ${\tilde{x}}_{m,j}=c_{j}\left(x_{m,j}-p\right)+p$ as defined in (Equation 10), where $c_{j}\in R$, j=1,…,k. Let
$$x_{m+1}=\arg\min_{j=1,\ldots ,k}||p-{\tilde{x}}_{m,j}\left|\right|.$$(iii)Let $G\left(x\right)=\cos\left(\theta_{x}\right)$, where x∈J, $\theta_{x}\in \left[0,\pi /2\right]$ is the acute angle of two vectors, $v_{x}=\left(p-x\right)$ and $n_{x}$, which denotes the normal vector to Ψ at x. If $1-G(x_{m+1})<\tau$, set $\hat{d}\left(p,H\right)=d\left(p,x_{m+1}\right)$ and stop, where $\hat{d}\left(p,H\right)$ denotes the estimate of d(p,H). Otherwise, set m=m+1 and repeat Steps (i) and (ii).

Figure 6 presents a schematic diagram of the FS algorithm at the *m*-th iteration. The idea of the FS algorithm comes from the gradient descent approach. In Step 1 of the algorithm, the initial point $x_{0}$ is determined by looking around the neighborhood of p and selecting the searching direction by $y_{r}$, which can be treated as a hint of the direction with high possibilities to reach an $x^{*}\in D_{p}$. Similarly, the algorithm selects the next searching direction around $x_{m}$ at the *m*-th iteration in the Step 2(i). This searching method mimics the scenario when one looks for a way out by a flashlight in a dark room, where *r* and δ in Steps 1 and 2(i), respectively, play the roles of different irradiation distances of the flashlight. Therefore, we name the above scheme by FS algorithm.

By using the FS algorithm in each iteration of the PSO procedure, the optimal solutions of the criteria (Equation 3) and (Equation 6) can be obtained with the boundary points $p_{i},i=1,\ldots ,n$. In the rest of this section, we derive the convergence of the FS algorithm. To begin, assume the following conditions.

C1.Ψ is continuously differentiable, denoted by $\Psi \in C^{1}$.C2.For any fixed $p\in R^{3}$, x∈J, and d(p,x)>d(p,H), there exists a 2-dimensional neighborhood $\kappa_{x}$ of x, where $\kappa_{x}=B_{\zeta }^{\left(2\right)}\left(x\right)⋂J$ with $B_{\zeta }^{\left(2\right)}\left(x\right)$ being a 2-dimensional open ball of radius ζ∈(0,ϵ] centered at x, such that $d\left(p,x\right)>d\left(p,\kappa_{x}\right)$ and for any $x_{a}$ and $x_{b}\in \kappa_{x}\cap J_{i}$, i=1,2,…, and for any α∈[0,1], Ψ(·) satisfies $\Psi \left(\alpha x_{a}+\left(1-\alpha \right)x_{b}\right)\Psi \left(p\right)\leq 0$ or $\Psi \left(\alpha x_{a}+\left(1-\alpha \right)x_{b}\right)\Psi \left(p\right)>0$.

The above two conditions are set on the function Ψ, which means that these two conditions can be checked for any selected family of shape equations. Specifically, C1 is a smoothness condition, which is satisfied by the equations with polynomial, exponential, logarithmic, and trigonometric components. C2 is a local convexity condition defined on a 2D subspace in $R^{3}$. An intuitive way to understand C2 is to cut a cashew, for example, with a knife and check whether C2 is satisfied by the shape of the boundary of the cross-section.

In this study, we employ the ellipse and cashew-shaped equations to depict the uptake regions in the 2D and 3D spaces, respectively. Apparently, the families of equations defined in (Equation 2) and (Equation 4) both satisfy C1. Next, we demonstrate that the family of ellipse equations satisfy C2. Without loss of generality, we consider the equation of the normal to the ellipse, that is, $K\left(x,y\right)=ax^{2}+by^{2}=1$, a,b>0. For any $\left(x_{1},y_{1}\right)\neq \left(x_{2},y_{2}\right)$ such that $K\left(x_{1},y_{1}\right)=K\left(x_{2},y_{2}\right)=1$, let $\left(u,v\right)=\alpha \left(x_{1},y_{1}\right)+\left(1-\alpha \right)\left(x_{2},y_{2}\right)$, where 0≤α≤1. By Jensen’s inequality and since $f\left(x\right)=x^{2}$ is a convex function, we have $K\left(u,v\right)\leq \alpha K\left(x_{1},y_{1}\right)+\left(1-\alpha \right)K\left(x_{2},y_{2}\right)=1,$ which implies that (u,v)∈{(x,y):K(x,y)≤1} for all 0≤α≤1 and, therefore, {(x,y):K(x,y)≤1} is a convex set. As a result, let Ψ(x)=K(x)−1 and if $\Psi \left(x_{a}\right)=\Psi \left(x_{b}\right)=0$, where $x_{a}\neq x_{b}$, we have $\Psi \left(\alpha x_{a}+\left(1-\alpha \right)x_{b}\right)\leq 0$ for α∈[0,1], and C2 holds no matter p is an interior or exterior point of {x:Ψ(x)=0}. For the cashew-shaped equations, C2 is also satisfied. However, the calculation is more tedious than that for the ellipse equations. We put the details in the Appendix A.

Based on C1 and C2, we have the following result.

**Theorem** **1.**
*Let $x_{m}$, m=0,1,2,…, be a sequence of points obtained from the FS algorithm. If C1 and C2 hold, then the sequence $d(p,x_{m})$ converges to d(p,H) as δ→0 and τ→0.*


The proof the Theorem 1 is given in the Appendix A. Most importantly, Theorem 1 indicates that the proposed FS algorithm is capable of computing an accurate distance from a fixed point to a given surface when the surface equation satisfies the smoothness and local convexity conditions as shown in C1 and C2, respectively.

## 4. Numerical Results

In this section, we introduce the details of extracting features from the ellipses and cashew-shaped surfaces. In addition, 6 popular classification methods are employed to investigate whether these features are useful for identifying PD subjects from the SPECT images. The SPECT images of 634 subjects with 305 female and an average age of 68.3 years old collected from Kaohsiung Chang Gung Memorial Hospital between January 2017 and May 2018 are employed for our investigation. For each subject, 3 SPECT images are selected and combined to a 2D-combined image, and 4 SPECT images are selected for fitting cashew-shaped surfaces according to physicians’ expertise and experience. Three experienced board-certified nuclear medicine physicians are invited to give their opinions on the 2D-combined image of each subject. Blinded to patient clinical symptoms and histories, the physicians determine the images independently according to the guideline. The final consensus result of each subject was assigned when at least 2 physicians achieved a common agreement, where 256 subjects were assigned to be abnormal (grade 2 or 3), 77 subjects were potentially abnormal (grade 1), 174 were normal, and 127 were nearly normal, which includes the slightly degenerated cases without neurodegenerative disorder of PD. For simplicity, we combine the first 2 categories as abnormal and the last two categories as normal, denoted by 1 and 0, respectively. Figure 2 shows examples of 2D-combined images for the above 2 labels.

Before collecting the shape features, a sensitivity study for the settings of ($c_{1},c_{2},\omega )$ in (Equation 8) is provided. In the literature, the parameters $c_{1}$, $c_{2}$, and *w* are set differently for different data [30,31,32]. In these studies, $c_{1}$ and $c_{2}$ are tuned between 1 and 3, and ω is tuned between 0.1 and 1.2. In the following, we conduct a sensitivity analysis with setting $c_{1}\in \left\{0.8,1.0,1.2\right\}$, $c_{2}\in \left\{1.6,2.0,2.4\right\}$, and ω∈{0.4,0.5,0.6} based on our experience when applying PSO to our data for the subjects shown in Figure 2 and Figure 7. Table 1 reports the results of these 27 different combinations of $\left(c_{1},c_{2},\omega \right)$. From the means and standard deviations presented in Table 1, we found that the values of $n^{-1}\sum_{i=1}^{n}d\left(p_{i},\Psi_{2D}\right)$ and $n^{-1}\sum_{i=1}^{n}d\left(p_{i},\Psi_{3D}\right)$, which are the values of the objective functions defined in (3) and (6), respectively, are not sensitive to the 27 different settings. To save the computational costs, we adopt $c_{1}=1$, $c_{2}=2$ and ω=0.5 in the following comparison study.

Once the parameters in Equations (3) and (6) are estimated by the convergent values of the proposed algorithm with the SPECT images for each subject, we treat these parameters of the ellipse and cashew-shaped equations as the features extracted from 2D and 3D ROIs, respectively. To investigate whether the features obtained from the ellipse and cashew-shaped equations are useful, we consider the following 6 different sets of features and compare their classification performances for PD identification.
Set 1Grayscale features (9 features): For each 2D-combined image, we compute the empirical density of the grayscale values, which are normalized into [0,1], and calculate 8 probabilities between two adjacent percentiles in {0.60,0.65,…,1} to represent the ratios of areas having different magnitudes of uptakes. In addition, we also compute the ratios of average uptake values near striatum to the average uptake values of the background.Set 2The features extracted from the ellipse equations (12 features): We employed the PSO-FS algorithm to approximate the shape of striatum for each 2D-combined image by the criterion defined in (Equation 3), where 100 particles are randomly set centered on an initial estimator of $\theta_{2D}$. Figure 2 presents approximated ellipses in red of a normal subject and a subject with PD, which reveal that the PSO-FS algorithm is capable of obtaining satisfactory ellipses to approximate the shapes of uptakes.Set 3The features extracted from the cashew-shaped equations (22 features): The procedure for fitting a 3D structure is similar to the above 2D shape fitting process. Figure 7 presents the sequences of 2D images in the upper panel and the corresponding cashew-shaped surfaces in 3D space in the lower panel of a normal subject and a subject with PD, where the 2D images in the red rectangles are recommended by the physicians and are used to construct the cashew-shaped surfaces. The fitting results shown in Figure 7 display that the PSO-FS algorithm is capable of obtaining satisfactory cashew-shaped equations to approximate the shapes of uptakes.Set 4The union of Sets 1 and 2.Set 5The union of Sets 1, 2, and 3.Set 6The 34 features defined in Table 1 of [8].

Next, 6 well-known classification methods, which are support vector machine (SVM), naive Bayes (NB), random forests (RF), XGboost (XGB), logistic regression (LR), and linear discriminant analysis (LDA), are used to learn classifiers for PD imaging classification with the above 6 sets of features separately [3,9,23,24,25]. The comparison study is conducted under the following repeated hold-out with 100 rounds of 5-fold cross-validation (CV) and stratified sampling scheme.
We randomly split the data into training and test sets with the proportions of 80% and 20%, respectively, under a stratified sampling scheme, where the proportions of normal, nearly normal, potentially abnormal, and abnormal subjects in the training and test sets are the same.In the training set, we conduct a stratified 5-fold CV framework on each of the SVM, NB, RF, XGB, LR, and LDA classifiers by R-packages svm(), naiveBayes(), randomForest(), xgboost(), glm(), and lda(), respectively. The tuning parameters in each of the SVM, RF, and XGB methods are determined by computing the validation accuracies under several settings of tuning parameters and selecting the one with the highest validation accuracy. For each of the SVM, RF, and XGB classifiers, we set 5 candidates for each tuning parameters centered at the default values of the corresponding R packages. Therefore, we have 6 candidate classifiers, where each of them has the best 5-fold CV performances for the training data.Let the classifier with the best 5-fold CV accuracy from the 6 candidates be our final selection. Learn this classifier again with all the training data and use it to compute the classification performances of the test data.Repeat the above 3 steps 100 times. Compute the mean and standard deviation (SD) of accuracy (ACC), sensitivity (SEN), specificity (SPE), and GM under the 5-fold CV framework of the 6 classifiers for training data, and compute the mean, SD, and 95% confidence interval (CI) of the 4 measurements for test data based on the results of the 100 rounds.

In the last step of the above procedure, the ACC, SEN, SPE, and GM [33] are defined by
$$\begin{array}{ccc} \mathrm{ACC} & = & (\mathrm{TP}+\mathrm{TN})/\mathrm{Total}\;\mathrm{population}\\ \mathrm{GM} & = & {\left(\mathrm{SEN}\times \mathrm{SPE}\right)}^{0.5}, \end{array}$$
where SEN=TP/Trueconditionpositive, SPE=TN/Trueconditionnegative, and ‘TP’ and ‘TN’ are short for ‘True Positive’ and ‘True Negative’, respectively.

The results of 5-fold CV and test sets are reported in Table 2 and Table 3, respectively. In Table 2, the highest AVEs of the ACC, SEN, SPE, and GM values among the 6 classifiers for each set of features are marked in bold, where the RF classifier has more robust performances than other classifiers for the 6 sets of features on the average. In Table 3, the highest AVEs of the ACC, SEN, SPE, and GM values among the 6 sets of features are marked in bold, where Set 5 has the highest ACC, SEN, and GM values. Interestingly, in the 100 rounds, the SVM and RF classifiers are selected to be our final classifier for test data most frequently with the features contained in Set 1. However, this is not the case for other sets of features. For example, the RF and LR classifiers are preferred for Set 3; the RF and LDA classifiers are preferred for Set 6. This phenomenon reveals that different sets of features or different data might need different classifiers to highlight their impact of classification, which is also the main reason why we use a 5-fold CV scheme to select our final classifier in step 3 of the above procedure.

In view of Table 2 and Table 3, we found that the classification performances did not depend on the number of features solely. For example, the ACCs and GMs of Set 1 are higher than those of Sets 2 and 3 in most cases although Set 1 contains less number of features than Set 2 or Set 3. Nevertheless, when we combine the features in Sets 1 and 2 into Set 4 or the features in Sets 1-3 into Set 5, the classification performances are improved significantly in terms of paired-wise *t*-tests as shown in the previous question. These phenomena reveal that the features in Set 1 did provide useful and important information for identifying PD but there is still room for improvement. The numerical results in Table 2 and Table 3 reveal that the 2D and 3D features in Sets 2 and 3, respectively, extracted from the proposed mathematical representations show their potential for improving the accuracy of automated PD identification.

Moreover, in Table 3, the average ACCs for test data based on 100 rounds with the 6 sets of features are 0.858, 0.792, 0.825, 0.873, 0.880, and 0.858, respectively. Since we use the same hold-out and 5-fold CV samples to learn classifiers with each of the 6 feature sets in each round, we adopt paired-wise *t*-tests on the ACCs of 4 cases, ‘Set 6 vs. Set 1’, ‘Set 1 vs. Set 4’, ‘Set 6 vs. Set 5’, and ‘Set 4 vs. Set 5’, to test whether these sets of features provide significantly different ACCs for test data. The *p*-values of these 4 cases are 0.44, $2.3\times 10^{-7}$, $4.3\times 10^{-11}$, and $7.3\times 10^{-4}$, respectively, which indicate that Set 5 ≻ Set 4 ≻ Set 6 ≈ Set 1, where A≻B denotes that the population mean of *A* is significantly higher than the population mean of *B*, and A≈B denotes that population means of *A* and *B* have no significant difference. Figure 8 presents the scatter plots of the ACCs of the 4 cases for test data based on 100 rounds, which reveal consistent phenomena with the statistical test results since most points in Figure 8b–d are located below the 45-degree line.

## 5. Discussion

A shape approximation approach is proposed to portray the ROI from medical images. An effective PSO-FS algorithm for obtaining a suitable approximation is proposed under the framework of PSO. The convergence of the PSO-FS algorithm is derived under mild assumptions on the selected family of equations. We apply the proposed procedure to the issue of detecting PD from the SPECT, where ellipse and cashew-shaped equations are selected to approximate the ROI in the 2D and 3D spaces, respectively. The coefficients of the ellipse and cashew-shaped equations, together with some well-known features of PD existing in the literature, are employed to learn PD classifiers under 6 commonly used machine learning approaches. The empirical results reveal that the coefficients of the ellipse and the cashew-shaped equations are capable of improving classification performances.

According to the findings in our empirical results, the proposed method has shown its potential to be applied to other medical imaging data to extract important features contained in the corresponding 2D and 3D images. Most importantly, once a suitable family of equations is selected to depict the regions of uptakes of a specific ROI based on the suggestions made by experts, the PSO-FS algorithm can help to obtain the optimal equation effectively if the selected family of equations satisfies some mild conditions. Therefore, more constructive and useful features can be computed according to the corresponding mathematical representations of the 2D and 3D shapes, which provide researchers the opportunities to gain more insights into the associated problems.

Moreover, although our empirical study provides numerical evidence to show the usefulness of the 2D and 3D shape approximations of the ROI for PD identification, the proposed method still has the following limitations:For a specific ROI, the proposed method needs to select a suitable family of mathematical representation, like the ellipse and cashew-shaped equations for identifying PD in 2D and 3D spaces, respectively. However, we may not have enough understanding or knowledge to derive useful features directly from a mathematical representation of any shape of the ROI, like the cashew-shaped equation used in this study. Therefore, we just adopt the coefficients of the shape equation as features for classification. Although these coefficients can represent the shape of the ROI, it may not be the most effective way for a suitable classifier to learn. More studies are needed to dig more insights for this issue.This study and [8] both adopted the family of ellipse equations to portray the ROI observed from an 2D-combined image for PD identification. However, the ROI for normal subjects should be comma-shaped, which is not an ellipse. Due to the reason that we know ellipses better than comma-shaped equations, the family of ellipse equations is selected to approximate the ROI. This selection, of course, has systematic biases (or called model risk), which might reduce the classification performance.In the proposed PSO-FS algorithm, we need to compute the distance of each boundary point of the ROI to a given shape equation in each PSO iteration. This procedure is computationally expensive, especially when training a classifier with a huge number of boundary points. A more efficiently computing way should be developed in the future.

## Figures and Tables

**Figure 1 sensors-20-05879-f001:**
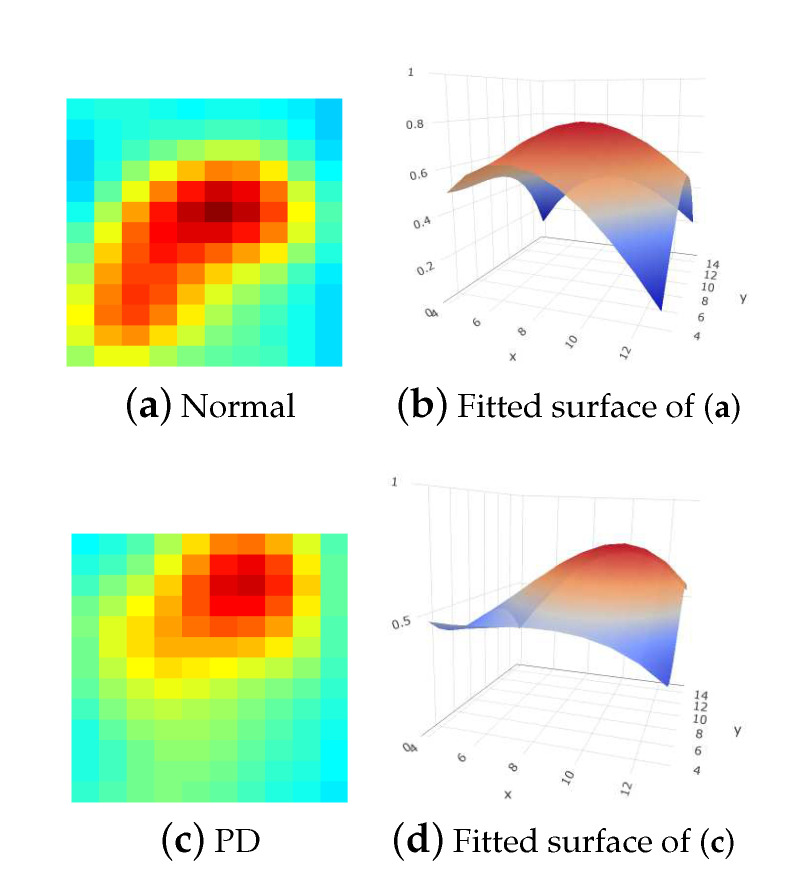
(**a**,**c**) are the 2D-combined SPECT images of a normal subject and a PD subject, respectively. (**b**,**d**) are the corresponding polynomial regression surfaces of order 3 for (**a**,**c**), respectively, obtained by Prashanth et al. [8]’s method.

**Figure 2 sensors-20-05879-f002:**
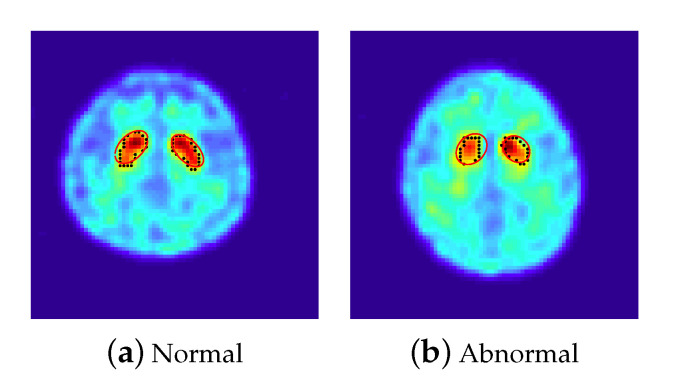
The boundary points, denoted by black points, and the approximated ellipses, denoted in red, of the region of uptakes in 2D-combined images for (**a**) a normal subject and (**b**) an abnormal subject.

**Figure 3 sensors-20-05879-f003:**
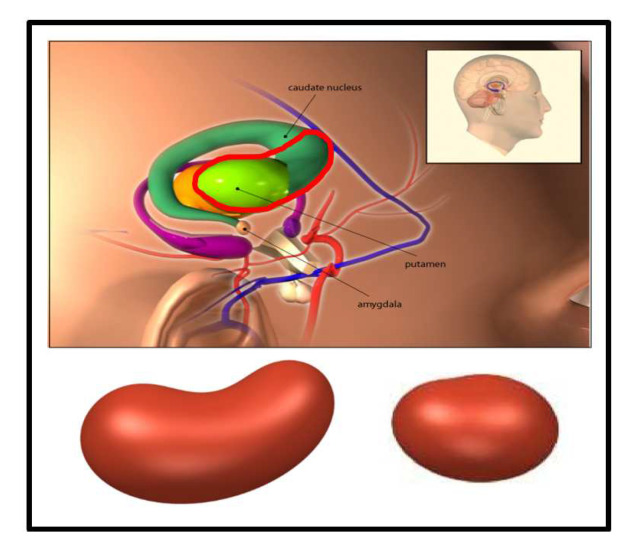
The 3D cashew-shaped equations to depict the ROI in 3D space, where the figure in the upper panel is downloaded from http://www.3d-brain.ki.se/atlas/images/basal_ganglia01.jpg, and the area marked with a red circle denotes the most active region of the highest-uptake. The left and right cashew-shaped structures shown in the lower panel are examples for a normal case and a PD case with two specific settings of the coefficients of the cashew-shaped equation defined in (Equation 4), respectively.

**Figure 4 sensors-20-05879-f004:**
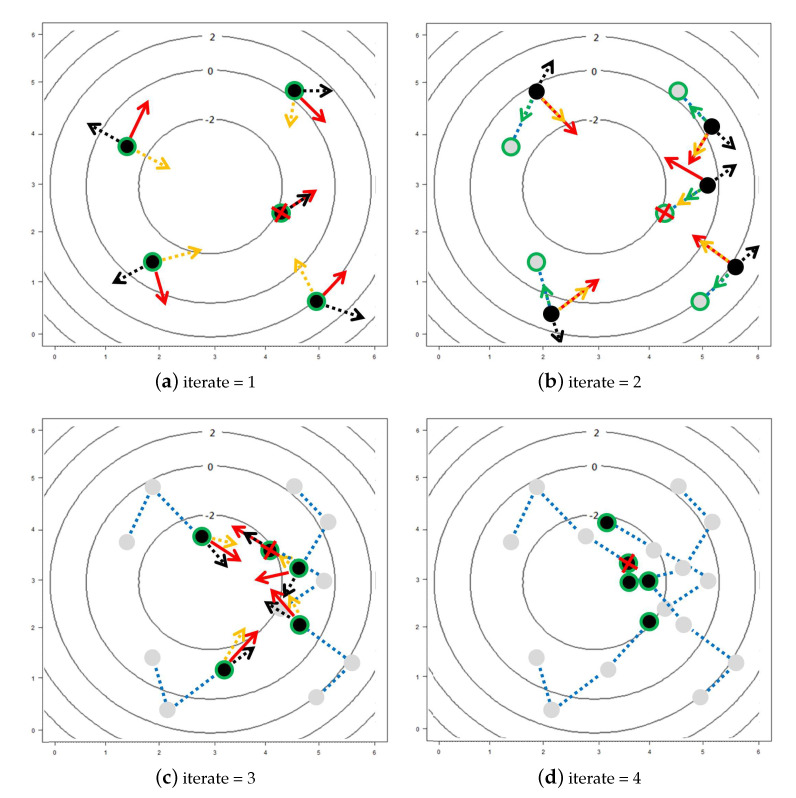
A schematic diagram of the Particle Swarm Optimization (PSO) algorithm with 5 particles in the first 4 iterations, where black points denote the current positions of the particles, gray points combined with dashed lines denote the corresponding path of each particle, and the red arrows denote the velocity of each particle determined by the black, green, and yellow dotted-arrows, which represent the three directions of the components on the right-hand side of (Equation 8). In the *t*-th iteration, the points with green circle denote $P_{i}^{t}$ for i=1,…,5, and the point with red cross denotes $G^{t}$, t=1,…,4, where $P_{i}^{t}$ and $G^{t}$ are defined in (Equation 8).

**Figure 5 sensors-20-05879-f005:**
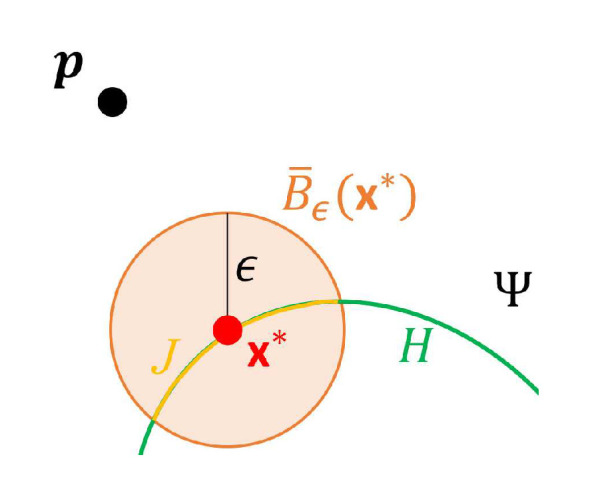
A schematic diagram of the flashlight-searching (FS) algorithm, where p is a point in $R^{d},d\geq 2$, *H* is the set of points satisfying Ψ=0, $x^{*}$ is the point such that $d\left(p,x^{*}\right)=d\left(p,H\right)$, ${\bar{B}}_{\epsilon }\left(x^{*}\right)=\{x:||x-x^{*}\left||\leq \epsilon \right\}$ denotes a closed ball of radius ϵ>0 centered at an $x^{*}$, and $J={\bar{B}}_{\epsilon }\left(x^{*}\right)⋂H$ denotes the arc of *H* associated with ${\bar{B}}_{\epsilon }\left(x^{*}\right)$.

**Figure 6 sensors-20-05879-f006:**
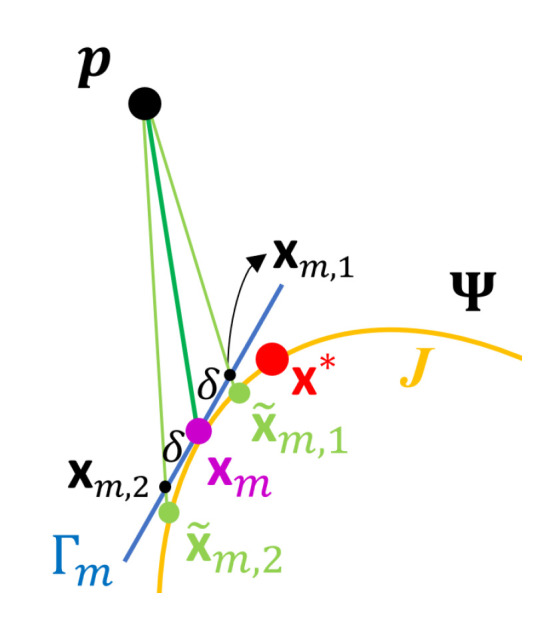
A schematic diagram of the *m*-th iteration of the FS algorithm, where p, Ψ, *J*, and $x^{*}$ are defined the same as in Figure 5, $x_{m}$ is the point at *m*-th iteration, $\Gamma_{m}$ is the tangent plane of Ψ at $x_{m}$, $x_{m,1}$, and $x_{m,2}$ are points satisfying $\left|\right|x_{m,j}-x_{m}\left|\right|=\delta$, δ is a predetermined positive constant, and ${\tilde{x}}_{m,j}$ is the points in *J* satisfying ${\tilde{x}}_{m,j}=c_{j}\left(x_{m,j}-p\right)+p$, and $c_{j}\in R$, j=1,…,k.

**Figure 7 sensors-20-05879-f007:**
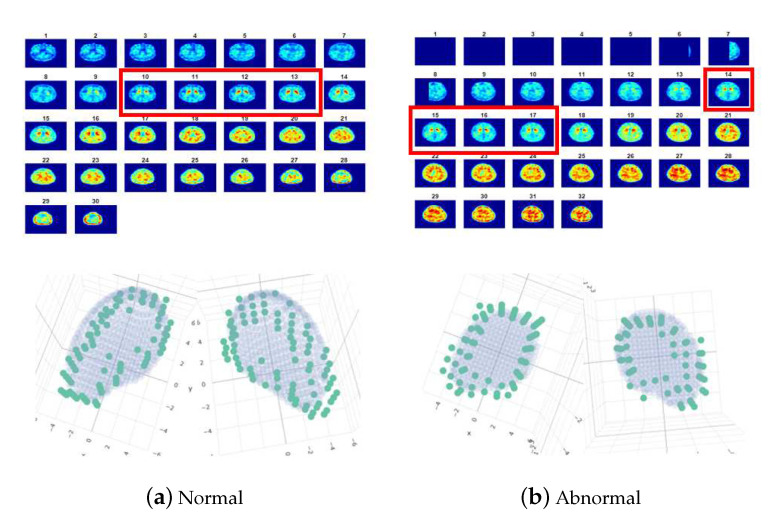
The sequences of 2D images (in the upper panel) and the corresponding cashew-shaped surfaces (in the lower panel) of (**a**) a normal subject and (**b**) an abnormal subject in 3D space. In the upper panel, the four 2D images in the red rectangles are suggested by the physicians and are used to construct the cashew-shaped surfaces. In the lower panel, the green points denote the boundary points collected from the four 2D images for left/right striatum in 3D space, and the gray areas are the corresponding fitted cashew-shaped surfaces.

**Figure 8 sensors-20-05879-f008:**
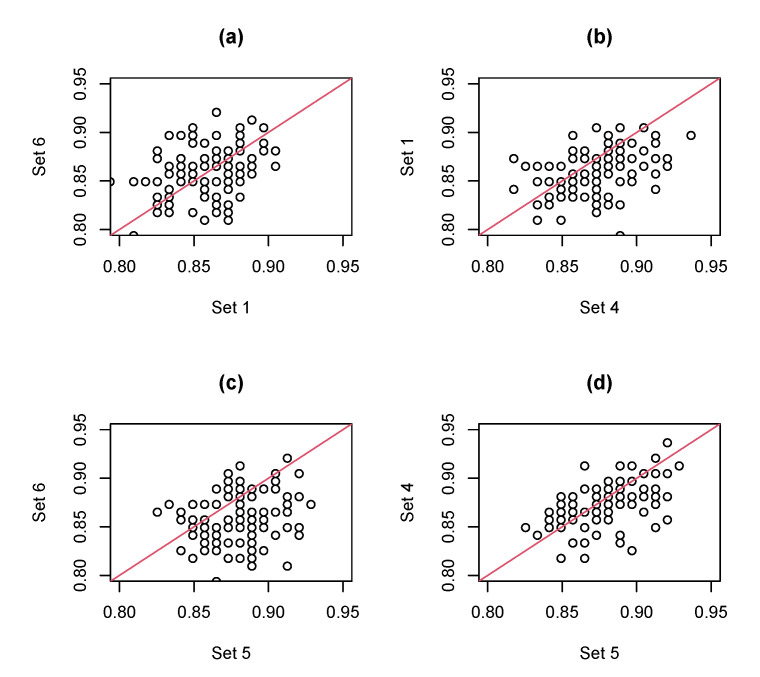
Scatter plots of the ACCs of different feature sets for test data based on 100 rounds: (**a**) Set 6 vs. Set 1, (**b**) Set 1 vs. Set 4, (**c**) Set 6 vs. Set 5, (**d**) Set 4 vs. Set 5, where the red line in each plot denotes the 45-degree line.

**Table 1 sensors-20-05879-t001:** The means and standard deviations (SDs) of $n^{-1}\sum_{i=1}^{n}d\left(p_{i},\Psi_{2D}\right)$ and $n^{-1}\sum_{i=1}^{n}d\left(p_{i},\Psi_{3D}\right)$ for the subjects in Figure 2 and Figure 7 based on 27 different settings of $\left(c_{1},c_{2},\omega \right)$ in (Equation 8), where $c_{1}=0.8,1.0,1.2$, $c_{2}=1.6,2.0,2.4$, and ω=0.4,0.5,0.6.

	Normal				Abnormal			
	2D		3D		2D		3D	
	L	R	L	R	L	R	L	R
mean	0.55	0.69	33.75	42.22	0.40	0.40	34.28	30.52
SD	0.01	0.14	1.43	3.81	0.01	0.04	0.97	0.95

**Table 2 sensors-20-05879-t002:** The AVEs and SDs of the ACC, SEN, SPE and GM values of the SVM, NB, RF, XGB, LR, and LDA classifiers with 6 collections of features for the 5-fold CV results based on 100 rounds, where the highest AVEs of the ACC, SEN, SPE, and GM values among the 6 classifiers for each set of features are marked in bold.

			Classifier					
Measure	Set		SVM	NB	RF	XGB	LR	LDA
ACC	1	AVE	0.865	0.753	**0.866**	0.836	0.842	0.848
		SD	0.008	0.008	0.009	0.013	0.009	0.008
	2	AVE	**0.798**	0.760	0.793	0.747	0.789	0.776
		SD	0.008	0.007	0.010	0.015	0.010	0.010
	3	AVE	0.798	0.805	**0.826**	0.788	0.824	0.819
		SD	0.012	0.009	0.010	0.016	0.010	0.009
	4	AVE	0.873	0.788	**0.880**	0.845	0.854	0.862
		SD	0.007	0.008	0.007	0.013	0.009	0.008
	5	AVE	0.824	0.824	**0.880**	0.835	0.855	0.859
		SD	0.010	0.008	0.007	0.012	0.011	0.009
	6	AVE	0.619	0.620	**0.865**	0.830	0.849	0.857
		SD	0.011	0.012	0.009	0.014	0.004	0.009
SEN	1	AVE	0.823	0.564	**0.846**	0.825	0.819	0.765
		SD	0.013	0.014	0.011	0.018	0.012	0.015
	2	AVE	0.760	0.642	**0.785**	0.734	0.772	0.700
		SD	0.015	0.011	0.013	0.018	0.013	0.016
	3	AVE	0.715	0.706	0.775	0.760	**0.799**	0.771
		SD	0.026	0.017	0.014	0.021	0.013	0.013
	4	AVE	0.845	0.656	**0.870**	0.828	0.836	0.802
		SD	0.010	0.013	0.008	0.017	0.013	0.011
	5	AVE	0.766	0.733	**0.874**	0.813	0.843	0.818
		SD	0.017	0.012	0.009	0.016	0.013	0.012
	6	AVE	**0.975**	0.286	0.852	0.815	0.846	0.812
		SD	0.006	0.024	0.010	0.017	0.007	0.013
SPE	1	AVE	0.911	**0.963**	0.888	0.848	0.867	0.939
		SD	0.014	0.005	0.012	0.019	0.013	0.010
	2	AVE	0.841	**0.891**	0.802	0.761	0.808	0.861
		SD	0.015	0.009	0.016	0.020	0.013	0.013
	3	AVE	0.891	**0.915**	0.882	0.818	0.853	0.873
		SD	0.020	0.011	0.012	0.019	0.014	0.012
	4	AVE	0.903	**0.935**	0.890	0.864	0.873	0.928
		SD	0.011	0.006	0.011	0.019	0.013	0.011
	5	AVE	0.888	**0.925**	0.888	0.858	0.869	0.904
		SD	0.016	0.009	0.009	0.018	0.014	0.011
	6	AVE	0.224	**0.990**	0.879	0.847	0.851	0.907
		SD	0.022	0.004	0.014	0.021	0.006	0.013
GM	1	AVE	0.866	0.737	**0.867**	0.837	0.843	0.848
		SD	0.008	0.010	0.009	0.013	0.009	0.008
	2	AVE	**0.799**	0.756	0.793	0.747	0.790	0.776
		SD	0.008	0.008	0.010	0.015	0.010	0.010
	3	AVE	0.798	0.804	**0.827**	0.788	0.825	0.820
		SD	0.012	0.010	0.010	0.016	0.010	0.009
	4	AVE	0.874	0.783	**0.880**	0.845	0.854	0.863
		SD	0.007	0.008	0.007	0.013	0.009	0.008
	5	AVE	0.825	0.824	**0.881**	0.835	0.856	0.860
		SD	0.010	0.008	0.007	0.012	0.011	0.009
	6	AVE	0.467	0.531	**0.865**	0.831	0.849	0.858
		SD	0.023	0.022	0.009	0.014	0.004	0.009

**Table 3 sensors-20-05879-t003:** The AVEs, SDs, and 95% CIs of the ACC, SEN, SPE, and GM values of 6 collections of features for the test data based on 100 rounds, where the highest AVEs of the ACC, SEN, SPE, and GM values among the 6 sets of features are marked in bold. In addition, $n_{1}$, $n_{2},\ldots ,$ and $n_{6}$, respectively, denote the numbers of times that the SVM, NB, RF, XGB, LR, and LDA classifiers being chosen in the 100 rounds.

		Measure	# Times
Set		ACC	SEN	SPE	GM	$n_{1}$	$n_{2}$	$n_{3}$	$n_{4}$	$n_{5}$	$n_{6}$
1	AVE	0.858	0.822	**0.898**	0.859	45	0	54	0	1	0
	SD	0.024	0.042	0.036	0.024						
	95%CI	(0.81,0.91)	(0.74,0.91)	(0.83,0.97)	(0.81,0.91)						
2	AVE	0.792	0.759	0.828	0.792	60	0	23	0	17	0
	SD	0.024	0.045	0.047	0.024						
	95%CI	(0.74,0.84)	(0.67,0.85)	(0.73,0.92)	(0.74,0.84)						
3	AVE	0.825	0.788	0.866	0.825	0	0	54	0	41	5
	SD	0.028	0.049	0.045	0.028						
	95%CI	(0.77,0.88)	(0.69,0.89)	(0.78,0.96)	(0.77,0.88)						
4	AVE	0.873	0.855	0.892	0.873	18	0	80	0	0	2
	SD	0.023	0.039	0.038	0.024						
	95%CI	(0.83,0.92)	(0.78,0.93)	(0.82,0.97)	(0.83,0.92)						
5	AVE	**0.880**	**0.872**	0.888	**0.880**	0	0	99	0	0	1
	SD	0.023	0.031	0.038	0.023						
	95%CI	(0.83,0.93)	(0.81,0.93)	(0.81,0.96)	(0.83,0.93)						
6	AVE	0.858	0.839	0.878	0.858	0	0	78	0	3	19
	SD	0.026	0.039	0.041	0.026						
	95%CI	(0.81,0.91)	(0.76,0.92)	(0.80,0.96)	(0.81,0.91)

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
