# Peer review of "A Shape Approximation for Medical Imaging Data"

_sensors, 2020, doi:10.3390/s20205879_

Round 1

Reviewer 1 Report

Abstract

It is required to provide critical quantitative indices of the classifier.

Also, the prevalence of the AD in the dataset and some demographics, such as the percentage of women and the average age, must be added.

Introduction

What is the gap in the literature?

Also, the literature review on AD diagnosis is missing.

Line 83

The equation number is missing.

Also, knowing that LS could be used to solve this equation and even equation 1, is it possible to follow convex optimization rather than stochastic optimization? Please discuss.

On the PSO algorithm,

How were the parameters c1, c2, and w were estimated? A sensitivity analysis is required.

Did the authors work on the convergence of the PSO, based on the values of PSO parameters?

Line 263,

How were the parameters of the classifiers tuned? Please discuss.

The hold out validation is used in this study, which is susceptible to Type III error. Either repeated hold-out or cross-validation must be used, and the mean and SD values over the test folds must be reported in Table 1. Moreover, following the STARD methodology, the CI 95% of the indices must be reported. Usually Type I and Type II errors are also reported. Please report such indices as well.

https://www.equator-network.org/reporting-guidelines/stard/

In Table 1, the highest accuracies were marked. It is necessary to use a proper statistical test to identify if a classifier significantly outperformed the others and not by chance.

The base classifiers LDA and NB must be added to Table 1, showing the complexity of the problem.

The discussion of the paper must be expanded toward the limitation of the algorithm, comparison with state of the art, and practical implications.

Reviewer 2 Report

The manuscript, which is related to an interesting approach to shape approximation for medical imaging data, is well prepared and presented. The optimization criteria adopted has a robust evaluation and its application in multiple domains seems interesting (e.g. 3D reconstruction). However, there are major concerns related to the literature review, model evaluation and discussion of results.

  • Introduction and related work

A critical analysis of the literature review is missing in the introduction. The authors only mentioned a few related works without clarifying the current limitations of these approaches. It is unclear why the authors are proposing a new methodology for a problem that is not discussed at the beginning of the manuscript. If the authors are proposing a robust feature extraction method, they should discuss the limitation of the SOTA methods.

Why the author did not consider the analysis of current methodologies based on deep learning techniques? Voxel-based morphometry is an automatic volumetric method employed for the detection of gray matter intensity reduction in the caudate and putamen regions. Deep learning neural network has recently emerged as a powerful analysis to exploit the spatial structure of subanatomical regions [1].

[1] Dolz J, Desrosiers C, Ayed IB (2017) 3D fully convolutional networks for subcortical segmentation in MRI: a large-scale study. NeuroImage 170:456–470

  • Experimental evaluation/model interpretation.

To compare the proposed methodology the authors only select a simple baseline that uses grayscale features (“Gray”). Why the authors did not compare their method with related works that were cited in the manuscript (Prashanth et al. 2016; Oliveira et al. 2018)?

It is noted that the authors aimed to demonstrate if the classification performance improved after adding the features extracted from the proposed methodology. However, it is unclear how was the classification performance by using only the 2D or 3D features. It is expected to have lower performance with just the “Gray” features due to the low number of features (9 features).

Why the authors adopted the proposed classifiers LR, NB, RF, SVM and XGB? Do these classifiers were used for other related works?

“Nevertheless, after adding the coefficients of the optimal ellipses and cashew-shaped equations into the 5 classifiers, we found significant improvement of the classification performances.” This is not actually true. Why the performance using the “Gray+2D+3D features” reduced with the SVM and LR classifiers. The model interpretation with the proposed features is missing. The authors should try to expand the discussion of the benefits of the proposed features by comparing with other feature engineering techniques.

How the authors demonstrate the statement related to the significance of the manuscript in the abstract with one single baseline method. “Our findings indicate that more constructive and useful features can be extracted from the proper mathematical representation of the 2D and 3D shapes…which helps to improve the accuracy of computer-aided diagnosis.” The authors should also evaluate a cross-validation scheme to test the generalisation of the proposed methodology with the different subject considered as the test set.

Page 10: “… the two classifiers are acceptable for the purpose of computer-aided diagnosis in practice.” This statement is highly ambitious because there is a huge difference between automated PD identification and clinical diagnosis.

Round 2

Reviewer 1 Report

The paper is now suitable for publication.

Author Response

Thank you for your comments. Your comments did enhance the quality of the manuscript.

Reviewer 2 Report

Thank you for addressing each concern. The manuscript has improved significantly. Just the changes in the methodology and experimental plan demonstrate a robust research that deserves publication. However, I still recommend minor changes for publication related to the literature review and discussion (please see attached document)
